# Overexpression of ErbB-1 (EGFR) Protein in Eutopic Endometrium of Infertile Women with Severe Ovarian Endometriosis during the 'Implantation Window' of Menstrual Cycle

**Jeevitha Poorasamy [1], Deepali Garg [2,\*], Juhi Bharti [2], Aruna Nambirajan [3], Asmita Patil [1], Jayasree Sengupta [1,†] and Debabrata Ghosh [1,†]**

[1] Department of Physiology, All India Institute of Medical Sciences, New Delhi 110029, India
[2] Department of Obstetrics and Gynecology, All India Institute of Medical Sciences, New Delhi 110029, India
[3] Department of Pathology, All India Institute of Medical Sciences, New Delhi 110029, India
[\*] Correspondence: drdeepaligarg@gmail.com
[†] Presently retired.

**Abstract:** The strong association between endometriosis and infertility is of high clinical significance. High proliferative bias in eutopic endometrium during the secretory phase is a hallmark of endometriosis, which may result in high occurrence of implantation failure and resultant infertility in endometriosis. The ErbB family of proteins regulates the proliferation capacity in the endometrium, potentially causing endometrial hostility to the implantation process in endometriosis. However, our knowledge regarding the involvement of the ErbB family in human endometrium during the window of implantation (WOI) in endometriosis-associated infertility is scant. In the present study, the cellular profiles of immunopositive ErbBs-1 to -4 in the endometrium of endometriosis-free, infertile women (Group 1; $n = 11$) and in eutopic endometrium of infertile women diagnosed with stage IV ovarian endometriosis (Group 2; $n = 13$) during the mid-secretory phase were compared using standardized guidelines. Computer-aided standardized combinative analysis of immunoprecipitation in different compartments revealed an overexpression of ErbB-1 in the epithelial, stromal and vascular compartments, along with marginally higher ErbB-3 expression ($p < 0.06$) in the vascular compartment and ErbB-4 expression ($p < 0.05$) in the glandular epithelium and stroma in the endometrium during the WOI in women with primary infertility associated with stage IV ovarian endometriosis compared with disease-free endometrium of control infertile women. It appears that changes in ErbBs in the eutopic endometrium during WOI induce anomalous proliferative, inflammatory and angiogenic activities in it, which can antagonize endometrial preparation for embryo implantation in endometriosis. This knowledge appears usable in strategizing methods for the treatment of endometriosis-associated infertility, as well as preempting the oncogenic potential of endometriosis.

**Keywords:** endometriosis; endometrium; ErbB receptors; immunoprecipitation; implantation stage; infertility

## 1. Introduction

The endometrium occupies a unique place in the human body as it undergoes a cyclical pattern of cellular growth and differentiation to accommodate the arrival of a fertilized embryo towards the establishment of pregnancy. According to the commonly accepted theory to explain the pathogenesis of endometriosis, the secretory phase of a non-fecund cycle ends with menstrual bleeding along with retrograde efflux of endometrial cells, some of which may adhere to organs within the peritoneal cavity [1]. Although such retrograde efflux of menstrual debris is a common event in women, it does not necessarily

lead to the development of endometriotic lesions [2]. In fact, one in ten women suffer from endometriosis, mostly while of reproductive age. Thus, it is possible that the primary defect exists in eutopic endometrium of women diagnosed with endometriosis, as it bears significant differences with control endometrium of endometriosis-free women [3,4]. There exists a strong association between endometriosis and infertility; in infertile women, the prevalence of endometriosis may be as high as 50%, and at least one in three women with endometriosis is infertile [5,6]. It appears that the pathophysiological basis of endometriosis-associated infertility is multifactorial, and that inadequate endometrial preparation causing implantation defect is an important issue [7]. Several groups have suggested that the proliferative capacity of the endometrium in women with endometriosis is higher than that compared with normal endometrium [8–13]. This endometrial anomaly may be a putative cause of the reportedly high occurrence of implantation failure in patients with endometriosis [14].

The epidermal growth factor receptor (EGFR) family mediates one important pathway in the regulation of proliferation in mammalian cells. This growth factor receptor family has four members: ErbB-1 (EGFR, HER1), ErbB-2 (HER2), ErbB-3 (HER3) and ErbB-4 (HER4). The ErbB family receptors are transmembrane glycoproteins with an extracellular ligand-binding domain, a transmembrane region and an intracellular domain displaying tyrosine kinase activity, except in ErbB-3. ErbB-1 and ErbB-4 form either homo- or heterodimers, while ErbB-2 functions as a cofactor for the other receptors. ErbB-3 needs obligatory heterodimerization because of its lack of tyrosine kinase activity. Receptor dimerization is essential for activation of the intracellular tyrosine kinase domain of ErbBs and phosphorylation of the C-terminal tail. Phosphotyrosine residues then activate, either directly or through adaptor proteins, downstream components of signaling pathways including Ras/MAPK, PLCγ1/PKC, PI (3) kinase/Akt, STAT and Par6-atypical PKC pathways [15–18].

A cyclical pattern of expression and localization of the ErbB family of receptor tyrosine kinases (RTKs) and their ligands in human endometrium has been reported [19]. In endometriosis, mRNAs for ErbB-1 and ErbB-3 are upregulated in eutopic endometrium of women with endometriosis [20–22]. To our knowledge, there is a lack of knowledge regarding the association between endometriosis and the expression of ErbB receptor family proteins in human endometrium during the window of implantation (WOI), i.e., days 20–24 of a typical 28-day menstrual cycle [23,24]. The accrued information, as summarized in Table 1, generally fails to provide any useful understanding in this regard, because the reported studies mostly failed to adhere to the WERF EPHect guidelines [25,26] and did not address the specific issue of cellular expression of ErbB family proteins in implantation-stage endometrium [19–22]. Our previous studies have pointed out that the physiology of secretory-phase endometrium is significantly affected in ovarian endometriosis-associated infertility [4,13,22]. Thus, the aim of the present study was to immunolocalize the cellular profiles of the ErbB family of proteins in endometrium of endometriosis-free, infertile women and in eutopic endometrium of infertile women diagnosed with severe (i.e., stage IV) ovarian endometriosis during the WOI. Per the WERF EPHect guidelines, sampling and data mining of tissue samples in the present study were conducted from only one subtype of endometriosis patients (i.e., severe ovarian endometriosis, stage IV), along with clear annotations. Thus, tissue samples collected during the mid-secretory phase from consenting patients diagnosed with primary infertility either without endometriosis or with diagnosed stage IV ovarian endometriosis and showing implantation-stage histological characteristics were used for immunohistochemical localization of ErbB receptor family proteins.

**Table 1.** Expression of ErbB family members in endometrium during endometriosis.

| Name of ErbB Family Member (Alias *) | Salient Observations [References] |
|---|---|
| ErbB-1 (EGFR, HER1) | Epithelial as well as stromal compartment of human endometrium express EGF and ErbB-1 [19,27]. In endometriosis, EGFR (ErbB-1) mRNA is upregulated in endometriotic eutopic endometrium, especially during the secretory period, compared with normal endometrium [20–22]. |
| ErbB-2 (HER2/neu) | ErbB-2 showed high expression during the early secretory phase [19]. Endometrial mRNA for ErbB-2 was higher in endometriotic eutopic endometrium compared with normal endometrium [20]. |
| ErbB-3 (HER3) | ErbB-3 showed high expression during the secretory phase [19,28]. Endometrial mRNA for ErbB-3 was higher in endometriotic eutopic endometrium compared with normal endometrium [20,22]. |
| ErbB-4 (HER4) | ErbB-4 showed high expression during the secretory phase [19,28]. Endometrial mRNA for ErbB-4 was comparable between endometriotic eutopic endometrium and normal endometrium [20]. |

* from the GeneCards homepage for the human gene database [29].

## 2. Material and Methods

### 2.1. Patient Selection and Tissue Processing

Patients enrolled in the Department of Obstetrics and Gynecology of the All India Institute of Medical Sciences-Delhi for surgical intervention for endometriosis and/or evaluation at the Infertility Clinic voluntarily participated in the study after understanding its purpose and providing written consent, according to the standard protocol. The study, approved by the Institutional Ethics Committee on the Use of Human Subjects (IECPG-546/21.10. 2020; RT-19/25.11.2020), was conducted according to the Declaration of Helsinki Amendment 2013. Infertile patients with primary infertility accompanied by stage IV ovarian endometriosis, classified as the patient group, or with no endometriosis, classified as the control group, were enrolled in the study as described elsewhere [4,13]. Thus, there were two groups and the sample size was calculated using Stata 14.0 (StataCorp LLC, TX, USA) and based on reported data [13] for power: 0.9 and $\alpha = 0.05$ with an attrition rate of 20% yielding $n = 11$/each group. In the initial recruitment, group 1 (control group) had seventeen (17) endometriosis-free patients and group 2 had twenty (20) patients diagnosed with stage IV ovarian endometriosis. For both groups, patients showing evidence of polycystic ovarian syndrome according to Rotterdam criteria [30], male factor of infertility and unexplained infertility were excluded. Confirmation of ovarian endometriosis and exclusion of other types of endometriosis were achieved from reports of pelvic imaging based on ultrasound, MRI and/or diagnostic laparoscopy. Severity stages and sub-type of the disease condition were defined according to rASRM protocol at the time of surgical laparoscopy and finally by histology, as described elsewhere [4,22,31]. None of the patients in the ovarian endometriosis group had prior clinical recording of the disease, and hence were not under any treatment for endometriosis. Exclusion criteria included the co-presence of any other endocrinological disorder, cancer, uterine conditions such as fibroids (leiomyoma) and adenomyosis, and tuberculosis, since these conditions might affect the results of the study, as described elsewhere [22,32]. The patients who had taken contraceptives, GnRH analogues, aromatase inhibitors, danazol, dienogest or anti-tuberculosis therapy during the past 6 months and/or who had undergone any previous laparoscopic surgery were not included.

Endometrial samples during the mid-ovulatory period were obtained from upper uterine fundus and collected in cold phosphate-buffered saline (PBS, pH 7.4) using a Karman cannula no. 4. Samples were immediately washed with cold PBS, transported to the laboratory on ice and immediately fixed with freshly prepared cold 4% (*w/v*) paraformaldehyde in phosphate buffer, processed and embedded in paraffin for histological assessment of endometrium. Histological assessment towards endometrial dating to identify implantation-stage endometrium was performed using hematoxylin–eosin-stained 5 µm paraffin sections according to the guidelines and previously optimized criteria of Noyes [33,34]. Based on endometrial dating independently performed by three investigators, 24 samples for both groups were finally identified as mid-secretory-phase endometrium (MSE), seen between cycle days 20 and 24 of a typical menstrual cycle of 28 days. These samples were used for immunohistochemistry, as described below. All chemicals were obtained from Sigma-Aldrich Inc. (St. Louis, MO, USA). A summary of the experimental design is shown in the form of a flowchart in Figure 1. Table 2 provides the synopsis of the samples used. Supplementary Table S1 provides the major clinical data for the selected patients in both groups.

**Table 2.** Summary of patients' characteristics.

| Parameter | Group | |
|---|---|---|
| | 1 | 2 |
| Group description | Control | OE-IV |
| Fertility history | Infertile * | Infertile * |
| Duration of infertility (months) | $21 \pm 9.0$ | $21.1 \pm 8.7$ |
| Recruited (Number) | 17 | 20 |
| Selected ** (Number) | 11 | 13 |
| Age in years [a] | $29.1 \pm 4.1$ | $29.8 \pm 4.7$ |
| BMI (kg/m$^2$) [a] | $20.4 \pm 3.2$ | $21.9 \pm 4.2$ |
| Length of menstrual cycle in days [a] | $28.7 \pm 1.3$ | $28.8 \pm 1.5$ |
| Cycle day of sample collection [a] | $22.6 \pm 2.3$ | $21.8 \pm 2.7$ |

* despite frequent and unprotected coitus. ** identified as mid-secretory-phase endometrium typical of WOI seen between days 20 and 24 of a typical ovulatory cycle using standardized endometrial dating procedure [33,34]. [a] for selected patients. Values are shown as means $\pm$ SDs.

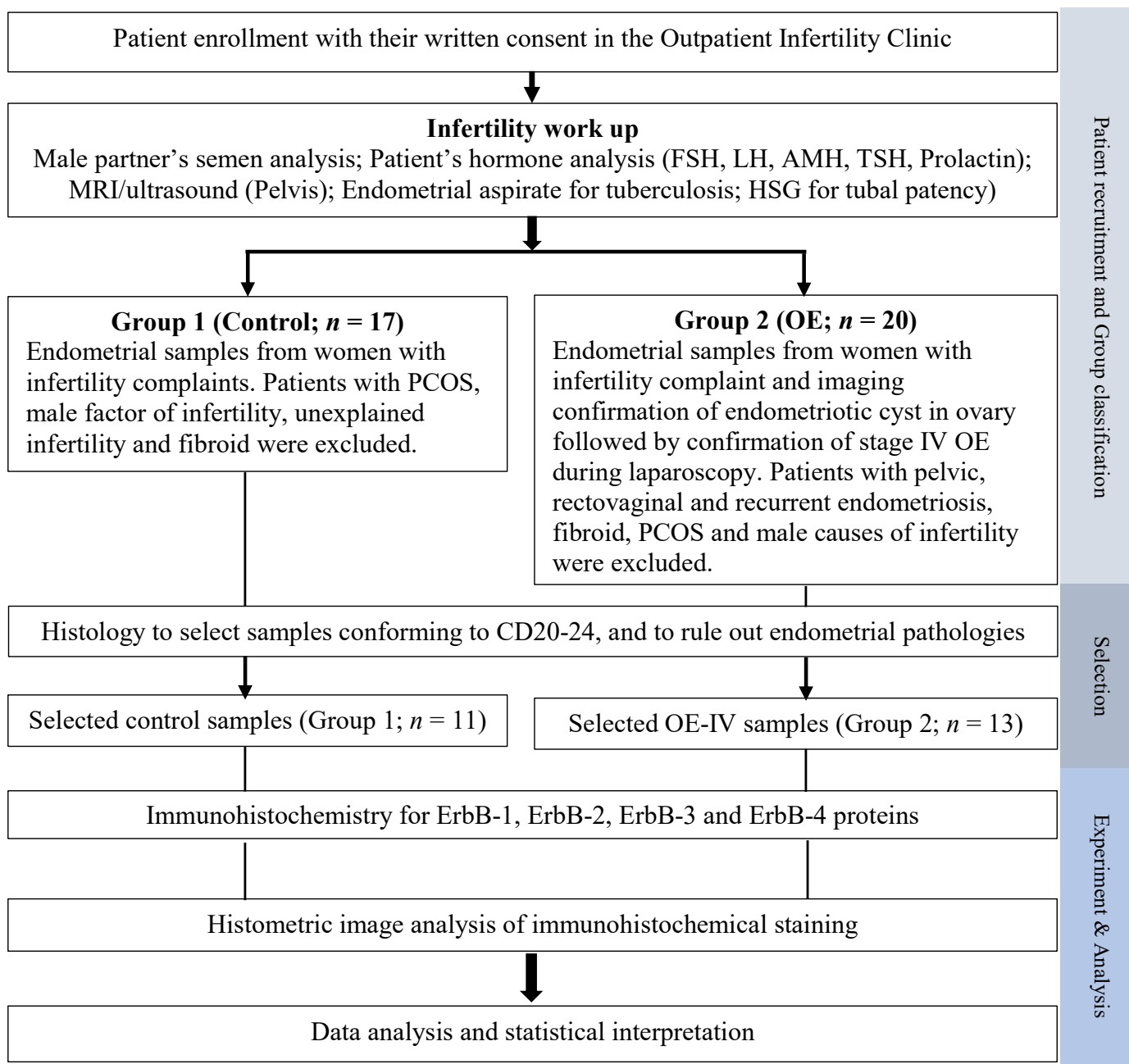

**Figure 1.** A flowchart of the overall experimental design used in the present study. AMH, anti-Mullerian hormone; CD, cycle day; FSH, follicular stimulating hormone; HSG, hystero-salpingogram; LH, luteinizing hormone; OE, ovarian endometriosis; OE-IV, stage IV ovarian endometriosis; PCOS, polycystic ovarian syndrome; TSH, thyroid-stimulating hormone.

*2.2. Immunohistochemistry*

Rabbit monoclonal antibodies against three antigens, namely ErbB-1, ErbB-2 and ErbB-3, and rabbit polyclonal antibody against antigen ErbB-4, obtained from Abcam (Cambridge, MA, USA), were used for immunohistochemistry (IHC) using 5 µm paraffin sections collected on poly-l-lysine-coated glass slides. Tissue sections were deparaffinized and subjected to heat retrieval in 0.1 M sodium citrate buffer (pH 6.1) for ErbB-1 and ErbB-4, and 0.5 M Tris-EDTA buffer (pH 9.0) for ErbB-2 and ErbB-3 using standard methods described elsewhere [35]. Briefly, endogenous peroxidase quenching was performed using 0.3% (*v/v*) freshly prepared hydrogen peroxide in phosphate buffer (pH 7.4), followed by blocking of non-specific binding using goat non-immune blocking serum (1:50) obtained

from Vector Laboratory (Burlingame, CA, USA). Dilutions of stock primary antibodies for incubation were pre-calibrated based on 5-point titration and the information provided by the manufacturer. The final visualization was achieved using the AEC-IHC kit (ab64260) obtained from Abcam (Cambridge, MA, USA) and Gill's hematoxylin obtained from Sigma-Aldrich (St. Louis, MO, USA). For IHC controls, sections were processed as above, with the omission of either primary antibody or secondary antibody.

*2.3. Image and Data Analysis*

All images were viewed, documented and analyzed using a Leica DMRD microscope and a Leica computer-assisted image analysis system (QWin DC 200) obtained from Leica Microsystems (Wetzlar GmbH, Germany). Three trained observers independently performed combinative scoring for immunostaining in different compartments for all samples. Staining vectors were digitally set using positive controls for spectral reference, which was applied in default mode individually to every slide to ensure meaningful detection and quantification of positive cells for every sample. The observers were blinded to the patients' clinical data during the scoring procedures. For every parameter, the optimal score in each compartment was assessed using a pre-calibrated, standardized five-point scoring scale that transformed continuous quantitative data into ordinal data [0 (<5%), 1 (5–25%), 2 (26–50%), 3 (51–75%), 4 (>75%)] according to the previously detailed procedure [35,36]. All scores provided by the observers were entered into a database using an Excel sheet and analyzed using weighted κ-statistics for the assessment of inter-observer errors yielding a mean κw-score of 0.67, suggestive of agreement beyond chance [37]. This approach of histometric analysis of immunohistochemical staining has been observed to be satisfactory and is recommended as an acceptable routine procedure in pathological studies [38,39]. Statistical analyses between two groups were performed using the Mann–Whitney *U*-test [40] using Graph Pad version 9 (GraphPad Software Inc., La Jolla, CA, USA) statistical packages.

## 3. Results

### 3.1. General

As seen in Table 2, 24 samples obtained from women during the mid-secretory phase of endometrial cycles from both groups were considered usable in the present study. These volunteers displayed similar physiological characteristics in terms of their reported duration of infertility, age at the time of tissue sample collection, and normal BMI and menstrual cycle length. Their tissue samples also displayed histology typical of the 'implantation window'. In the following section, the profiles of immunopositive ErbB family proteins in selected endometrial samples from control (group 1) and stage IV ovarian endometriosis (group 2) groups are presented. A composite plate showing representative photomicrographs at low magnification for ErbB immunostaining in MSE obtained from both groups is provided in Supplementary Figure S1.

### 3.2. Immunohistochemical Localization

ErbB-1: Figure 2 displays the representative photomicrographs of ErbB-1 immunostaining in different compartments of MSE obtained from both groups and the ordinal scores in respective compartments. In the control samples, ErbB-1 immunoprecipitation was seen in basal and apical regions of luminal epithelial cells with sparse immunostaining in glandular epithelial cells and minimal membrane and cytoplasmic staining within stromal cells of subluminal zone and in stromal cells surrounding glands of zone III (Figure 2A,B). Eutopic endometrial samples obtained from endometriosis patients displayed higher ErbB-1 immunoprecipitation in cell cytoplasm, cell borders and nuclei of luminal epithelia in basal and apical regions, and nuclei of glandular epithelial cells, in stromal cells of the subluminal compartment, and in cells surrounding glands in zone III (Figure 2D,E). A marked difference was observed in the ErbB-1 immunopositivity in the vascular compartment of eutopic endometrial samples from endometriosis patients compared with the control (Figure 2G–J). Minimal ErbB-1 immunopositivity was detected in endothelial cells

of spiral arterioles and surrounding pericytes in control endometrium (Figure 2G), while significant immunopositivity was detected in the vascular endothelium and in surrounding pericytes of eutopic endometrium as well as in the blood-borne cells within blood vessels and in the extracellular matrix surrounding glands of eutopic endometrium (Figure 2H–J). These features were generally absent in control endometrium. The histometric scoring for ErbB-1 protein in epithelial, stromal and vascular compartments of endometrial samples revealed significantly higher immunoprecipitation in all four compartments (viz., luminal epithelium: $p < 0.01$; glandular epithelium: $p < 0.01$; stroma: $p < 0.02$; endothelium: $p < 0.02$; vascular pericytes: $p < 0.01$) of functionalis of eutopic endometrium obtained from women with primary infertility and ovarian endometriosis (group 2) as compared to endometrium obtained from women in the primary infertility endometriosis-free control group (group 1) (Figure 2K–M).

ErbB-2: Figure 3 displays the representative photomicrographs of ErbB-2 immunostaining in different compartments in MSE obtained from both groups (Figure 3A–J) and the scores in respective compartments. Immunohistochemical localization of ErbB-2 in endometrium showed basal and apical distribution of immunoprecipitates in luminal and glandular epithelium, often around nuclear regions, along with its widespread presence in stromal cells, particularly in the subluminal zone and around the glands. There were no marked differences in samples between control women with primary infertility without endometriosis (group 1) and with stage IV ovarian endometriosis (group 2). No difference was seen in semi-quantitative scoring of immunopositivity detected for ErbB-2 protein in epithelial, stromal and vascular compartments of endometrial samples obtained from eutopic endometrium of women with primary infertility as compared to control endometrial samples from endometriosis-free women with primary infertility (Figure 3G–I).

ErbB-3: Figure 4 displays the representative photomicrographs of ErbB-3 immunostaining in different compartments in MSE obtained from both groups and the scores in the respective compartments. Figure 4A–J shows representative photomicrography of immunopositivity detected for ErbB-3 protein in epithelial, stromal and vascular compartments of control and eutopic endometria of women with primary infertility and without or with severe ovarian endometriosis. In control endometrium, marked ErbB-3 immunostaining was detected in the basal and apical regions of epithelial cells (Figure 4A,B), along with minimal ErbB-3 immunopositive staining in vascular endothelial cells and pericytes (Figure 4G). A similar profile of immunopositivity for ErbB-3 was detected in eutopic endometria in epithelial and stromal compartments (Figure 4D,E). Although a marginally higher ($p < 0.06$) expression of ErbB-3 was seen in the vascular cells of eutopic endometrium from severe ovarian endometriosis, it was statistically not different from control values (Figure 4M).

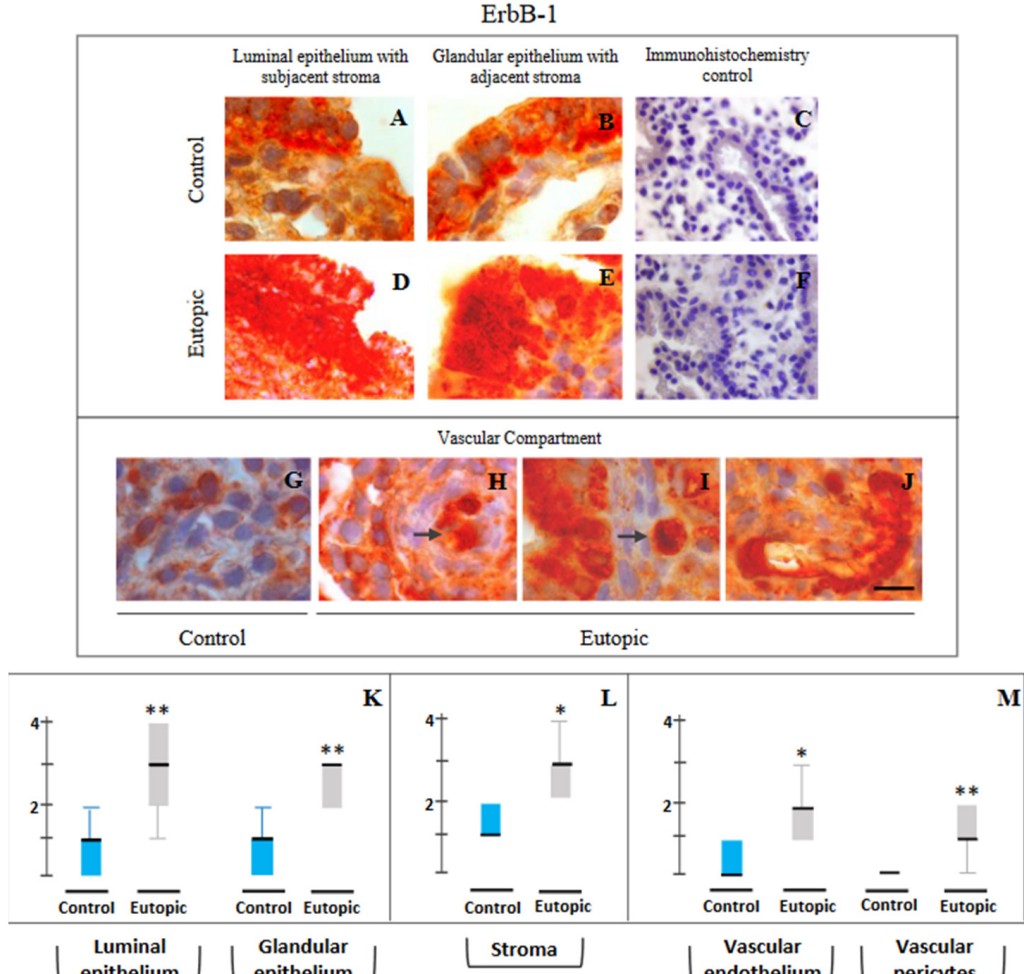

**Figure 2.** Representative photomicrographs of ErbB-1 (**A–J**) expressions in luminal epithelium along with subjacent stroma (**A,D**), glandular epithelium along with adjacent stroma (**B,E**) and in cells of the vascular compartment (**G–J**) in endometrial samples collected from endometriosis-free control women of group 1 (**A–C,G**) and stage IV ovarian endometriosis patients of group 2 (**D–F,H–J**). Semi-quantitative assessments of immunopositivity scores are shown in the form of box plots of 10–90 percentile distribution of scores, along with median values (**K–M**). Markedly higher ErbB-1 immunostaining is seen in the luminal and glandular epithelium and in stromal cells of endometrium obtained from group 2 subjects, i.e., infertile patients with stage IV ovarian endometriosis (**D,E**), as compared to control endometrium obtained (**A,B**) from endometriosis-free control women (group 1). High levels of cytoplasmic distribution besides membranous localization of ErbB-1 are seen, more predominantly in eutopic endometrial epithelium (**D,E**). Minimal immunopositivity is seen in endothelial cells and pericytes of control endometrium (**G**), while endothelial cells and pericytes of eutopic endometrium obtained from patients with severe endometriosis show strong immunostaining for ErbB-1 (**H–J**). Intravascular (**H**) and extravascular (**I**) monocytes in eutopic endometrium showed moderate to high immunoprecipitation (shown as arrows). This feature was generally absent in control endometrium obtained from disease-free control patients. Marked cytoplasmic and nuclear distribution besides membranous localization are notable for ErbB1 expression in eutopic endometrium of stage IV ovarian endometriosis infertile patients. Controls for immunohistochemistry staining were attained by omitting the primary antibody (**C**) or the secondary antibody (**F**). * $p < 0.02$. ** $p < 0.01$. Bar: 20 μm (**A,B,D,E,G–J**), 100 μm (**C,F**).

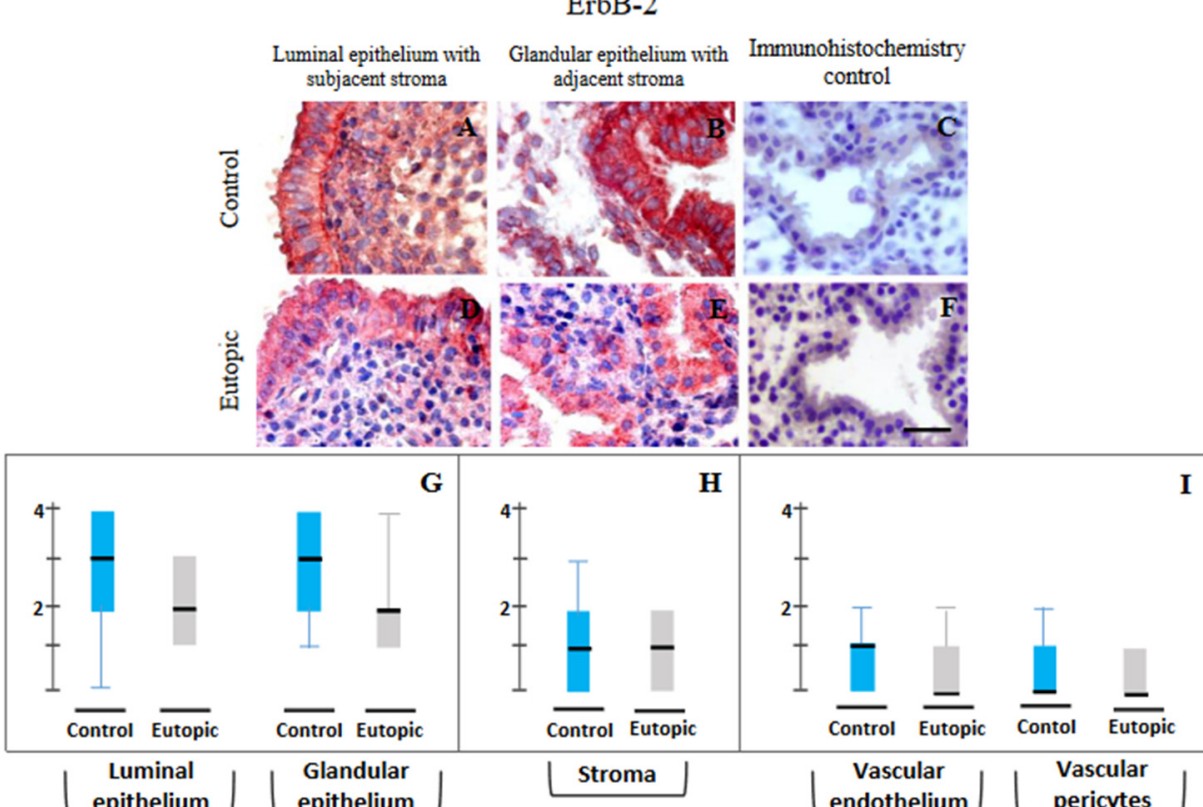

**Figure 3.** Representative photomicrographs of ErbB-2 (**A–F**) expressions in luminal epithelium along with subjacent stroma (**A,D**), glandular epithelium along with adjacent stroma (**B,E**) in endometrial samples collected from endometriosis-free control women of group 1 (**A–C**) and stage IV ovarian endometriosis patients of group 2 (**D–F**) and their semi-quantitative assessments shown in the form of box plots of 10–90 percentile distribution of scores, along with median values (**G–I**). No marked difference was observed in any compartment between the two groups. Controls for immunohisto-chemistry staining were attained by omitting the primary antibody (**C**) or the secondary antibody (**F**). Bar: 75 μm.

ErbB-4: Figure 5A–F documents the representative photomicrography of ErbB-4 immunopositivity in the epithelial and the stromal compartments of endometriosis-free control endometrium (Figure 4A,B) and eutopic endometrium from ovarian endometriosis patients (Figure 5D,E). Semi-quantitative scores of immunopositivity detected for ErbB-4 protein in endometrial samples (Figure 5G–I) obtained from control women (group 1) and patients with stage IV ovarian endometriosis (group 2) revealed only a marginally ($p < 0.05$) higher trend of ErbB-4 immunopositivity in the glandular epithelium and stromal components of eutopic endometrium as compared to control endometrium (Figure 5G,H).

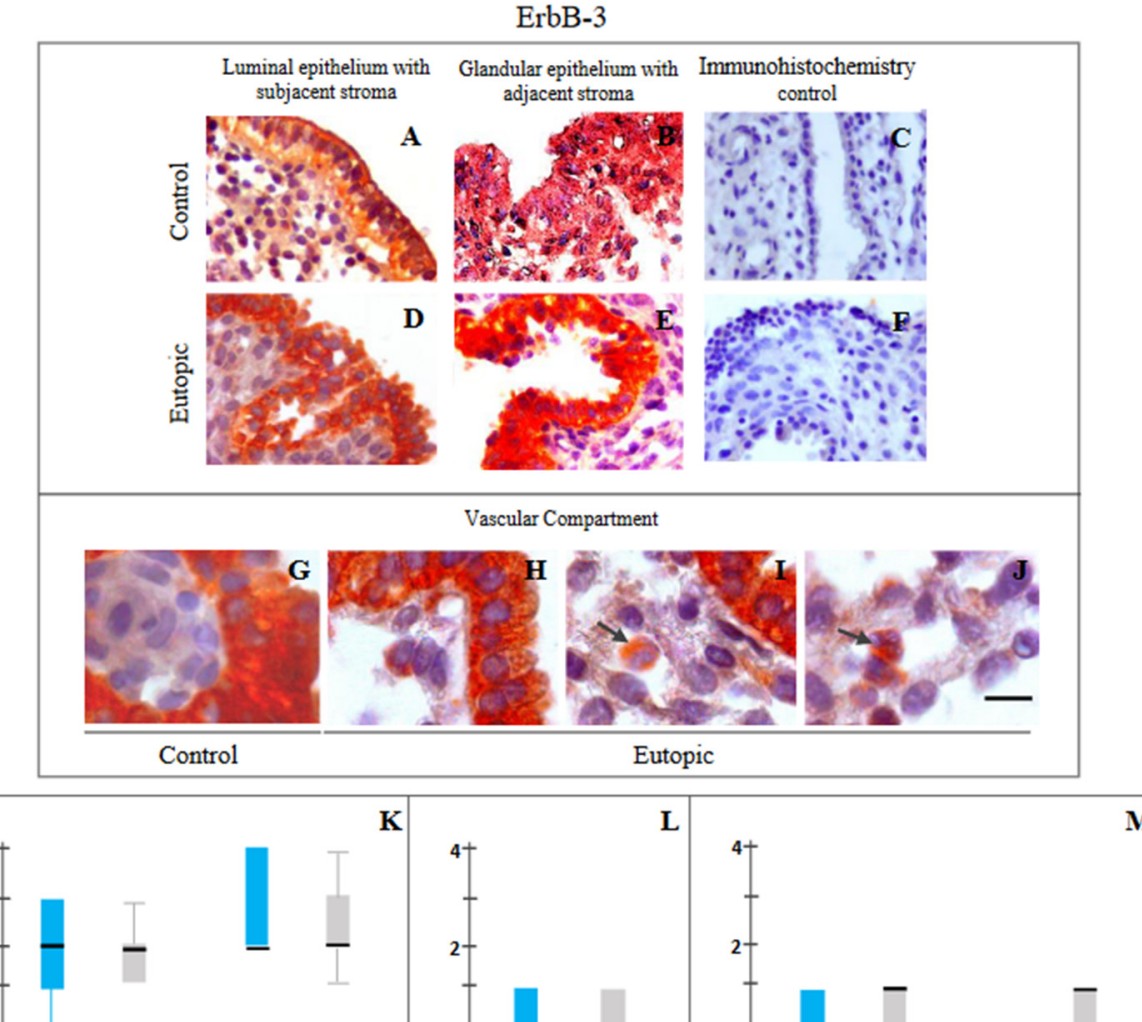

**Figure 4.** Representative photomicrographs of ErbB-3 (**A–J**) expressions in luminal epithelium along with subjacent stroma (**A,D**), glandular epithelium along with adjacent stroma (**B,E**) and in cells of the vascular compartment (**G–J**) in endometrial samples collected from endometriosis-free control women of group 1 (**A–C,G**) and stage IV ovarian endometriosis patients of group 2 (**D–F,H–J**). Semi-quantitative assessments of immunopositivity scores are shown in the form of box plots of 10–90 percentile distribution of scores, along with median values (**K–M**). Despite apparent higher expression in the vascular compartment of eutopic endometrium from ovarian endometriosis group, it was not statistically significant. Intravascular monocytes in eutopic endometrium showed moderate to high immunoprecipitation (shown as *arrows*). This feature was generally absent in control endometrium obtained from disease-free control patients. Marked cytoplasmic and nuclear distribution besides membranous localization are notable for ErbB3 immunoexpression in eutopic endometrium of stage IV ovarian endometriosis infertile patients. Controls for immunohistochemistry staining were attained by omitting the primary antibody (**C**) or the secondary antibody (**F**). Bar: 20 μm (**G–J**), 75 mm (**A,B,D,E**), 100 μm (**C,F**).

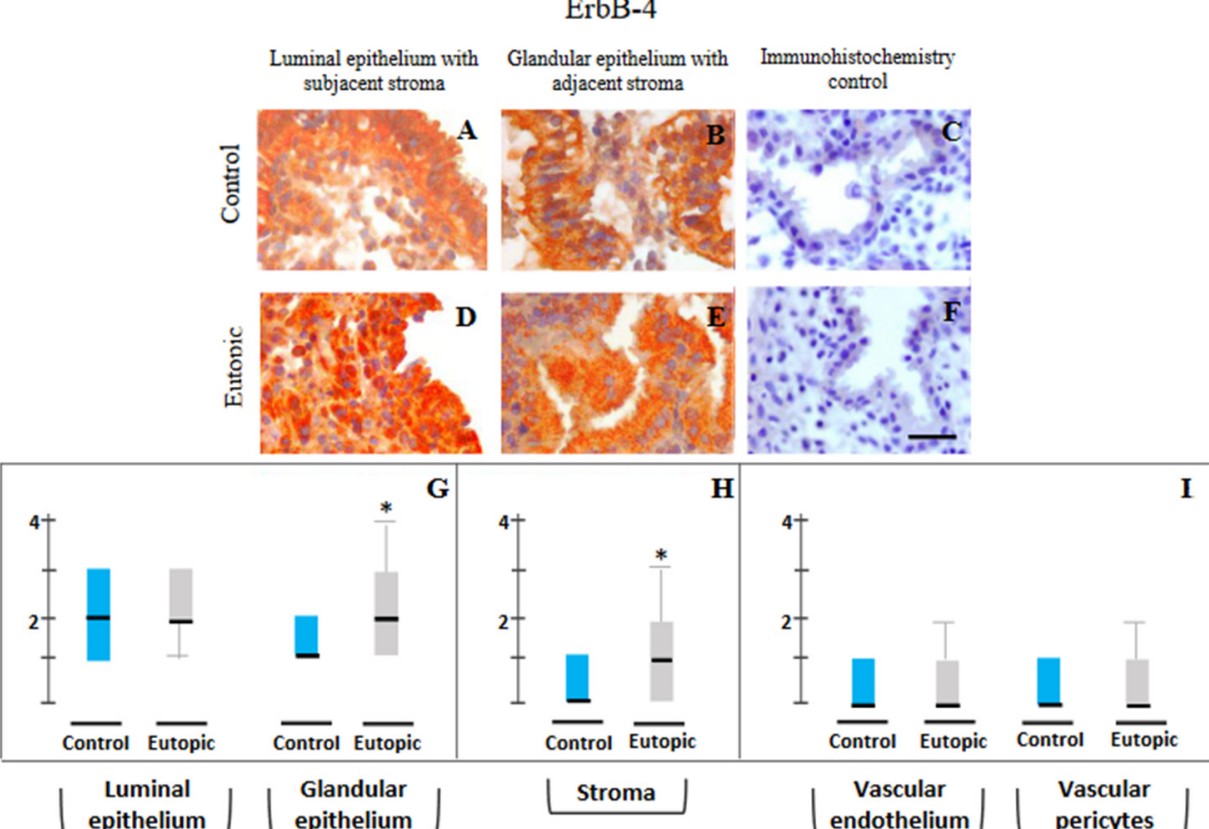

**Figure 5.** Representative photomicrographs of ErbB-4 (**A–F**) expressions in luminal epithelium along with subjacent stroma (**A,D**), and glandular epithelium (**B,E**) in endometrial samples collected from endometriosis-free control women of group 1 (**A–C**) and stage IV ovarian endometriosis patients of group 2 (**D–F**) and their semi-quantitative assessments shown in the form of box plots of 10–90 percentile distribution of scores, along with median values (**G–I**). High levels of cytoplasmic distribution besides membranous localization of ErbB-4 are seen, more predominantly in eutopic endometrial epithelium (**D,E**). A marginally higher trend of ErbB-4 immunopositivity is seen in the glandular epithelium and stromal components of eutopic endometrium as compared to control endometrium (**G,H**). Controls for immunohistochemistry staining were attained by omitting the primary antibody (**C**) or the secondary antibody (**F**). * $p < 0.05$. Bar: 75 µm (**A,B,D,E**), 100 µm (**C,F**).

## 4. Discussion

This is the first report of differential expressions of the ErbB family of tyrosine kinase receptors in mid-secretory-phase endometrium (MSE) obtained from endometriosis-free women diagnosed with primary infertility versus patients with primary infertility and severe ovarian endometriosis. We observed an overexpression of ErbB-1 (EGFR/HER1) in the epithelial, stromal and vascular compartments, along with marginally higher ErbB-3 expressions in the vascular compartment and ErbB-4 expression in the glandular epithelium and stroma in endometrium functionalis during the 'window of implantation' of women with severe (stage IV) ovarian endometriosis compared with control endometrium. The results obtained in the present study appear clinically useful, revealing the specific issue of association between endometrial expression of ErbBs during the implantation stage and infertility associated with severe ovarian endometriosis per the rASRM guidelines.

ErbB-1 (EGFR) apparently plays an integral role in establishing the cellular framework necessary for a successful pregnancy [41]. In a normal ovulatory cycle, ErbB-1 controls proliferative events, while ErbB-2, ErbB-3 and ErbB-4 influence the secretory maturation of endometrium [19,28,42–44]. There are previous reports indicating that the relative expressions of the ErbB family may markedly vary in eutopic endometrium with severe

endometriosis from control endometrium obtained from endometriosis-free infertile patients [20–22]. It is generally known that ErbB-1 (EGFR) plays an important role in cell proliferation and that secretory-phase endometrium from women with endometriosis displays a proliferative molecular profile with an enrichment of genes involved in cell cycle regulation [45–47]. Additionally, it has been reported that ErbB-1/EGFR signaling may result in aberrant cAMP-induced in vitro decidualization of stromal cells in women with endometriosis via cooperation between EGFR and protein kinase A signaling in the regulation of PI3K/AKT/mTOR [48–50]. EGFR signaling pathways, leading to altered in vitro responses to steroid hormones by endometrial stromal cells of endometriosis, are different from normal endometrial cells [51]. Interestingly, in a mouse model of endometriosis, ErbB-1/EGFR-mediated ERK1 and activator protein 1 signaling for the transcriptional activation of MMP-7 in epithelial cells was observed, and the treatment with an EGFR inhibitor led to the regression of endometriotic lesions along with decreased MMP-7 activities [52]. Taken together, it appears that overexpression of ErbB proteins, especially ErbB-1 in epithelial, stromal and vascular compartments in the implantation-stage endometrium obtained from patients with severe endometriosis, may cause endometrial hostility and failure of embryo implantation due to the hyper-proliferative status in eutopic endometrium during severe ovarian endometriosis [8–13].

Additionally, the observation in the present study that both ErbB-1 and ErbB-3 were overexpressed in the vascular compartment of eutopic endometrium of women with severe-stage endometriosis is a matter of interest. Peripheral blood monocytes express ErbB-1, ErbB-2 and ErbB-3 on their cell surfaces [53,54]. An inhibitory effect of ErbB-3 on the proinflammatory activation of CD14$^{low}$CD16$^{+}$ monocytes that show marked adherence to endothelial cells was earlier reported [54]. ErbB-1 signaling is known to regulate macrophage function via EGFR signaling-activated NF-κB and MAPK1/3 pathways to induce cytokine production and macrophage activation [55]. Thus, overexpressed ErbBs may cause observed 'hyperinflammatory bias' during the implantation window in eutopic endometrium with severe ovarian endometriosis [13].

It is notable in this connection that co-expression of EGF and EGFR in the secretory phase of the normal menstrual cycle, coupled with co-expressions of VEGF, FGF and their respective receptors, coincides with the timing of the development of sub-epithelial capillary plexus [43,56]. Thus, higher expression of ErbB-1 in the endothelium and pericytes may result in angiogenic phenotypes in eutopic endometrium during severe ovarian endometriosis [57–59]. Furthermore, we have noted an apparent similarity of ErbB-1 expression between a WHO grade IV EGFR-amplified glioblastoma sample and eutopic endometrium obtained from patients with severe endometriosis (Figure 6). Such an expression pattern of ErbB-1 induces higher proliferative capacity, increased vessel density, cellular atypias, high mitotic activity and distinctive infiltrative phenotype in both types of tissues, and these changes may bring forth their oncogenic potential [57,60–65]. The observed cytoplasmic and nuclear distribution of ErbBs, especially for ErbB-1 and ErbB-3, besides their membranous localization in the eutopic endometrium during endometriosis, may trigger the pathogenic potential of eutopic endometrium [66–70]. Collectively, it appears from the results of the present study that there was an overexpression of ErbB-1 in endometrial epithelium, stromal and vascular cells during the implantation phase, which might explain how endometrial preparation for embryo implantation could be disturbed due to anomalous proliferative, inflammatory and angiogenic activities in the target tissues in severe ovarian endometriosis, resulting in associated infertility.

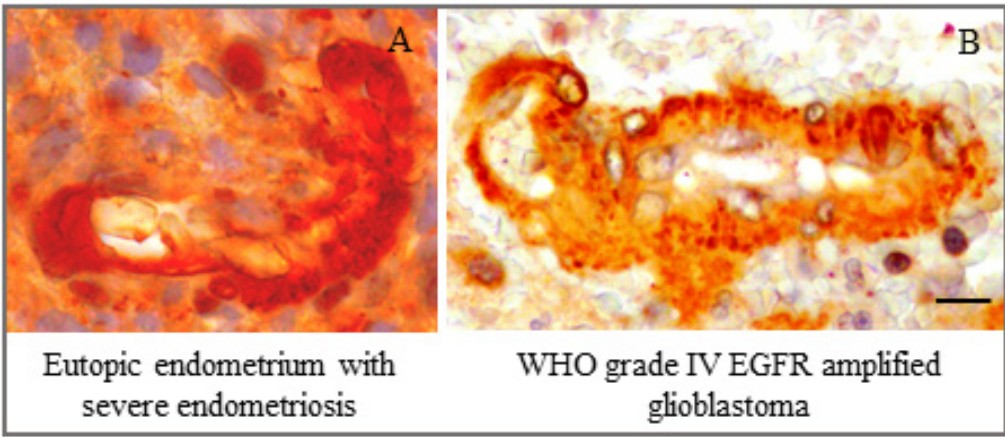

**Figure 6.** Marked similarity of ErbB-1 expression in the vascular compartment between (**A**) eutopic endometrium obtained on cycle day 24 from a woman with severe ovarian endometriosis-associated infertility and (**B**) a WHO grade IV EGFR-amplified glioblastoma sample. Bar: 10 μm.

There were a few limitations in the present study. Firstly, we could recruit only a limited number of subjects due to the stringent application of WERF EPHect guidelines [25,26], and due to the histological criteria for identifying mid-secretory phase having features of window of implantation, as previously defined [33,34]. Additional ultrasound investigation of follicular rupture, which would provide solid support to the data of histological dating in individual patients, could not be performed given the outpatient set-up of the present study. Despite these limitations, we believe that the results of the present study are indeed useful due to the stringent administration of the EPHect model and the endometrial dating model for tissue selection. Secondly, there was no design to undertake any functional studies towards understanding the specific roles of ErbB-1 in epithelial, stromal and vascular cell types during severe-stage ovarian endometriosis. Further investigations to interrogate the roles of ErbB-1 in these cell types and that of ErbB-1 and ErbB-3 in monocytic cells in eutopic endometrium in the disease state are necessary to address these limitations. In the present study, the issue of endometrioma-related reduction in ovarian reserve (ERROR) was not explored [71,72]. Thus, the question of whether the overexpression of ErbB-1 in endometrium of infertile patients with endometriosis could be a consequence of altered endocrine milieu, particularly estrogen action, which is known to influence the regulation of ErbBs, could not be addressed [73–75]. This could identify the association, if any, between these two factors, namely estrogen receptors and ErbB-1, in endometrium of infertile patients during endometriosis. In this connection, it is notable that Miturski et al. failed to obtain any correlation between ErbB-1 and estrogen receptor expressions in endometrial carcinomas [76]. However, a higher level of tissue estrogen in eutopic endometrium of the ovarian endometriosis group compared to the non-endometriosis group of infertile patients in the secretory phase of the menstrual cycle was observed in our previous study [4]. Furthermore, we have also proposed an association between estrogen and progesterone receptor subtypes in eutopic endometrium of infertile women with ovarian endometriosis based on our reported data that may lead to increased cell proliferation, cell migration, decidual incompetence and inflammatory responses leading to failure of embryo implantation [4,10,77]. We now report an added factor in this scenario: increased cellular (membrane and cytoplasmic) expressions of ErbB-1 in glandular, stromal and inflammatory cells of eutopic endometrium. Further study to link stage IV ovarian endometriosis with expression of the ErbB family of proteins and associated molecular pathways, as well as to unravel the functional association between stage IV ovarian endometriosis and ErbB family expression in endometriosis-associated infertility will strengthen our understanding and yield improved modes of treatment and management of this disease. Lastly, parallel investigations on an additional control group of normal women with proven fertility donating endometrial samples during mid-secretory-phase receptivity would yield a higher order of

knowledge; however, this was not possible in the outpatient hospital set-up of the present study. Future studies using alternative experimental models, e.g., primary cell culture and cell lines, may help in filling some of the gaps in this knowledge domain.

In conclusion, a preferential and accentuated expression of ErbB-1 in all compartments of endometrium *functionalis* during the critical 'window of implantation' in women with severe ovarian endometriosis and infertility appears novel and intriguing. This knowledge can be of help in strategizing methods for the treatment of patients with endometriosis and infertility, as well as preempting the oncogenic potential of endometriosis.

**Supplementary Materials:** The following supporting information can be downloaded at: https://www.mdpi.com/article/10.3390/reprodmed3040022/s1, Figure S1: Composite plate of representative photomicrographs at low magnification for ErbBs1-4 immunostaining in MSE obtained from both groups; Table S1: Characteristics of subjects and endometrial biopsies in the study.

**Author Contributions:** Conceptualization, J.S. and D.G. (Debabrata Ghosh); methodology, J.P., D.G. (Deepali Garg), J.B., A.N., J.S. and D.G. (Debabrata Ghosh); software, J.P. and D.G. (Debabrata Ghosh); validation, J.P., A.N., J.S. and D.G. (Debabrata Ghosh); formal analysis, J.P., D.G. (Debabrata Ghosh); investigation, D.G. (Deepali Garg), J.B., A.N. and A.P.; resources, D.G. (Deepali Garg), J.B., A.N. and A.P.; data curation, J.P., J.S. and D.G. (Debabrata Ghosh); writing—original draft preparation, J.S.; writing—review and editing, J.P., J.B. and D.G. (Debabrata Ghosh); visualization, J.P., J.S. and D.G. (Debabrata Ghosh); supervision, D.G. (Debabrata Ghosh); project administration, D.G. (Deepali Garg), J.B. and A.P.; funding acquisition, J.B. and D.G. (Deepali Garg). All authors have read and agreed to the published version of the manuscript.

**Funding:** This research was funded by All India Institute of Medical Sciences, grant numbers F.8.A-829/2020/RS and F.8-736/A-736/2019/RS.

**Institutional Review Board Statement:** The study was conducted in accordance with the Declaration of Helsinki, and approved by the Institutional Ethics Committee on the Use of Human Subjects (IECPG-546/21.10. 2020; RT-19, dated 25 November 2020).

**Informed Consent Statement:** Informed consent was obtained from all subjects involved in the study.

**Data Availability Statement:** Not applicable.

**Acknowledgments:** The authors acknowledge the support of facilities of the Cell and Molecular Physiology Laboratory of the All India Institute Medical Sciences, New Delhi. The authors express their gratitude to the patients who volunteered to participate after understanding the goal of the proposed study. The authors also express their gratitude to the reviewers for their comments. Funding support and tissue samples were received from project no. F.8.A-829/2020/RS (Deepali Garg.) and project no. F.8-736/A-736/2019/RS (J.B.).

**Conflicts of Interest:** The authors declare no conflict of interest.

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
