# Peer review of "Overexpression of ErbB-1 (EGFR) Protein in Eutopic Endometrium of Infertile Women with Severe Ovarian Endometriosis during the ‘Implantation Window’ of Menstrual Cycle"

_2673-3897, doi:10.3390/reprodmed3040022_

Round 1
Reviewer 1 Report (Previous Reviewer 1)
Authors have addressed most of the concerns raised before.
Author Response
We thank the Reviewer for the acknowledgement and acceptance of the revision incorporated based on the comments given.

Reviewer 2 Report (New Reviewer)
dear authors
thank you for submitting your work to this journal
we have read the submitted manuscript
we have a few suggestions
1.the abstract needs to be revised; endometriosis's role as an important cause of infertility needs to be mentioned; your recommendation at the end of the abstract or a clear home tacking massage needs to be added
2. introduction
the way that endometriosis (E) contributes to infertility needs to be discussed; kindly see [ Endometriosis can it cause infertility? ]
the defect in knowledge in literature needs to be addressed;2 published papers by your own institute is published; you did not discuss them nor the rationale for choosing this topic
3. Methods
it must be improved; Reproducibility is vital in clinical research.
no clear inclusion criteria for the controls? the 2 references that are mentioned do not explain how [infertile controls were chosen] did you exclude PCOS Cases? did you confirm tubal patency? did you check for male SFA? what about unexplained infertility were there any? Please add a flow chart to see the sequential exclusion of cases . Kindly add a sample size calculation to your methods.
4. results well described
5. Discussion many of your results were discussed without highlighting the clinical implication; in fact, most of your discussion led to an open end. We need to give the reader a clear home-tacking massage and how the current study will change our understanding .
7. References must be improved
more than 60% of them are 20 years old ;plears update
Author Response
Responses to the comments of the Reviewer 2 (shown in blue highlight)
(1) The abstract needs to be revised; endometriosis's role as an important cause of
infertility needs to be mentioned; your recommendation at the end of the abstract or a
clear home tacking massage needs to be added.
We thank the Reviewer for pointing out these hiatus. We have revised the Abstract section in
the present Re-revised manuscript per the suggestions made. Please see Page 2, lined 25-28
and 41-45.
(2) Introduction:
(i) The way that endometriosis (E) contributes to infertility needs to be discussed;
kindly see [Endometriosis can it cause infertility?]. The defect in knowledge in
literature needs to be addressed.
We have addressed this issue in the Introduction section of the present Re-revised
manuscript. Please see Page 3, lines 58-62, and reference 7.
(ii) Published papers by your own institute is published; you did not discuss them nor
the rationale for choosing this topic.
We have addressed this issue in the Introduction section of the present Re-revised
manuscript. Please see Page 4, lines 85-90.
(3) Methods:
(i) It must be improved; Reproducibility is vital in clinical research. No clear
inclusion criteria for the controls? the 2 references that are mentioned do not
explain how [infertile controls were chosen] did you exclude PCOS Cases?
did you confirm tubal patency? did you check for male SFA? what about
unexplained infertility were there any?
We have now included the asked details in the present re-revised manuscript. Please see Page
6, lines 119-121 and Figure 1.
(ii) Please add a flow chart to see the sequential exclusion of cases.
We have included the asked flowchart in the present re-revised manuscript. Please see Page
9, line 176 and Figure 1.
(iii) Kindly add a sample size calculation to your methods.
We have now included the asked details in the present re-revised manuscript. Please see Page
7, Table 2, lines 153-154.
(iv) Results well described.
Thanks for the acceptance of the Results section after revision.
(v) Discussion many of your results were discussed without highlighting the
clinical implication; in fact, most of your discussion led to an open end. We
need to give the reader a clear home-tacking massage and how the current
study will change our understanding.
We have revised the entire Discussion section in the present Re-revised manuscript per the
suggestions made. Please see Page 18, lines 371-374; Page 19, lines 392-396, lines 404-406
and lines 410-411; Page 20, lines 414-432; Page 22, lines 472-476.
3
| (vi) | References must be improved: more than 60% of them are 20 years old, please update. |
While revising the References, we felt that the old references as pointed out by the reviewer
are reports of original studies and we cannot delete them for good publication ethics. As the
next best step, we have added a few new and relatively recent references (please see
References 5-7, 49, 56, 58 and 67) along with the old references to make the list up-to-date in
the present re-revised manuscript.

Round 2
Reviewer 2 Report (New Reviewer)
dear authors thank you for accepting our sugestion
the paper aim and impact has improved
1.kindly see table 2, the numbers and terms in the third rows are deviated please make them in oneline
2. there is no formal calculation of the study sample size, the study is not powered you may add this as study limitation
This manuscript is a resubmission of an earlier submission. The following is a list of the peer review reports and author responses from that submission.
Round 1
Reviewer 1 Report
1) It is confusing as women with primary infertility are addressed as normal healthy women, which is clearly not a case and is misleading.
2) Ideal control would be endometrium collected from women with proven fertility during WOR but I understand getting these samples if very difficult, therefore alternate experiments should be planned using cell lines.
3) Quantitation can easily be re-done using software such as image J thus minimizing human induced error.
4) In fig 2 some of the images seem to be at higher magnification as compared to others ex: image O and R. Please make sure all are at same magnification.
5) Also provide images at lower magnification for a complete overview in the supplement data.
6) Please redo figure panels by splitting them to include quantitation and IHC for each group (ErbB-1/2/3/4) side by side.
7) Although it is acceptable to date endometrium histologically, it is more reliable to document follicular rupture using ultrasound to determine WOR.
Reviewer 2 Report
In “ Overexpression of ErbB-1 (EGFR) protein in eutopic endome-trium of infertile women with severe ovarian endometriosis during the ‘implantation window’ of menstrual cycle” Jeevitha Poorasamy et al. showed that the overexpression of ErbB-1 in all cell types in the endometrium during implantation window induces anomalous proliferative, inflammatory and angiogenic activities in women with severe ovarian endometriosis and hinders endometrial
preparation for embryo implantation.
The manuscript is interesting and well written.

Reviewer 3 Report
None
Reviewer 4 Report
This is a well-written article and an interesting research study shedding light to the mechanisms underneath implantation failure. Materials and methods are appropriate, results are well presented and conclusions are supported by the results. I support the manuscript's acceptance for publication.